# Burden of Disease Due to Ambient Particulate Matter in Germany—Explaining the Differences in the Available Estimates

**DOI:** 10.3390/ijerph192013197

**Published:** 2022-10-13

**Authors:** Myriam Tobollik, Sarah Kienzler, Christian Schuster, Dirk Wintermeyer, Dietrich Plass

**Affiliations:** 1German Environment Agency, Department Environmental Hygiene, Corrensplatz, 14195 Berlin, Germany; 2Berlin-Brandenburg Academy of Sciences and Humanities, Transfer Unit Science Communication, 10117 Berlin, Germany

**Keywords:** air pollution, particulate matter, environmental burden of disease, disability-adjusted life year, Germany, life table, attributable deaths

## Abstract

Ambient particulate matter (PM_2.5_) pollution is an important threat to human health. The aim of this study is to estimate the environmental burden of disease (EBD) for the German population associated with PM_2.5_ exposure in Germany for the years 2010 until 2018. The EBD method was used to quantify relevant indicators, e.g., disability-adjusted life years (DALYs), and the life table approach was used to estimate the reduction in life expectancy caused by long-term PM_2.5_ exposure. The impact of varying assumptions and input data was assessed. From 2010 to 2018 in Germany, the annual population-weighted PM_2.5_ concentration declined from 13.7 to 10.8 µg/m^3^. The estimates of annual PM_2.5_-attributable DALYs for all disease outcomes showed a downward trend. In 2018, the highest EBD was estimated for ischemic heart disease (101.776; 95% uncertainty interval (UI) 62,713–145,644), followed by lung cancer (60,843; 95% UI 43,380–79,379). The estimates for Germany differ from those provided by other institutions. This is mainly related to considerable differences in the input data, the use of a specific German national life expectancy and the selected relative risks. A transparent description of input data, computational steps, and assumptions is essential to explain differing results of EBD studies to improve methodological credibility and trust in the results. Furthermore, the different calculated indicators should be explained and interpreted with caution.

## 1. Introduction

Ambient particulate matter pollution is a serious threat to human health worldwide [1,2]. Especially for the small fraction of particles with an aerodynamic diameter less than 2.5 µm (PM_2.5_), the evidence base for long-term health effects is well-established. Studies continuously show associations with health outcomes even at very low concentration levels that are well below existing European limit values [3].

Burden of disease estimates are increasingly used to quantify the effects of PM_2.5_ on population health [4]. Currently, three institutions provide regular updates on the burden of disease attributable to PM_2.5_—the European Environment Agency (EEA), the World Health Organization (WHO), and the Institute for Health Metrics and Evaluation (IHME) [5,6,7].

Despite the predominant use of the comparative risk assessment representing the state-of-the-art method in this area, the resulting estimates often vary widely because of different input data and selected assumptions. For Germany, as an example, the EEA reports 63,100 deaths and 710,900 years of life lost because of mortality (YLLs) attributable to PM_2.5_ in 2018 [5]. For the same year, the IHME’s Global Burden of Disease (GBD) study presents 26,592 (uncertainty interval, 95% UI 19,743–33,900) deaths and 430,491 (95% UI 423,932–562,365) YLLs attributable to PM_2.5_. The most recent WHO estimates are only available for 2016, with 588,242 YLLs attributable to PM_2.5_.

Decisionmakers show an increased interest not only to have information on the overall number of YLLs or attributable deaths but also on the average loss in life expectancy (LE) that is due to PM_2.5_. Global assessments of life expectancy reductions indicated a loss of about one year on a global average and of around 0.4 years in Germany for 2016 [8]. Lelieveld and Pozzer [9] estimated a loss of 2.41 years; however, their estimate included the effects of ozone as well and they used a different exposure-response function (ERF).

The increasing number of available environmental burden of disease (EBD) estimates raises the question about which estimates are most reliable, valid, or simply most suitable for the intended purpose [4,10]. For such a comparison, it is important to also compare the input data and underlying assumptions used in the respective assessments.

The aim of this study is to compare different available EBD estimates using Germany as an example. Accordingly, for our calculation of estimates in the main analysis, we use input data that were specific for the situation in Germany. We present a time series of disease burden attributable to PM_2.5_ for the years 2010 to 2018. Furthermore, we use selected methodological variations for estimating the attributable burden and, in addition, calculate estimates for the reduction in life expectancy for comparisons with recent estimates from other institutions. Finally, we present and discuss differences of the estimates and reflect on the main factors causing these differences.

## 2. Materials and Methods

### 2.1. Quantification Method

In our main analysis, we used the WHO’s environmental burden of disease (EBD) approach to estimate PM_2.5_-related population health effects for five different health outcomes and the years 2010 to 2018 [11]. Thus, YLLs, years lived with disability (YLDs), disability-adjusted life years (DALYs), and the number of attributable deaths were quantified. Furthermore, we calculated the EBD using different calculation approaches to compare the main analysis results (see Table 1).

For our main analysis we have chosen health outcomes with the strongest scientific evidence base regarding the health effects due to PM_2.5_: chronic obstructive pulmonary disease (COPD), stroke, ischemic heart disease (IHD), tracheal, bronchus and lung cancer (LC), and type 2 diabetes mellitus (T2DM) [12], as done in the GBD-2019 study. All indicators were calculated for the years 2010 to 2018 using a common counterfactual value of 4.2 µg/m^3^, which was based on the same epidemiological studies from which the relative risks (RR) were taken. This counterfactual value was also used in the GBD-2019 study. We applied the same RR for mortality and morbidity effects, as done in the GBD-2019 study. The UI of the RR were used to estimate the UI for the burden of disease estimates. Uniform age weights and no time discounts were applied. The estimation processes were performed stratified by sex and five-year age groups. We used Microsoft Excel (version 2013) for all calculations. A sample spreadsheet can be found in the Appendix A. These results were compared with estimates from the GBD-2019 study [12].

Within method A, we used the AirQ+ software (version 2.07, WHO Regional Office for Europe, European Centre for Environment and Health (ECEH), Bonn, Germany) as provided by the WHO [13] to estimate the burden caused by the outcomes available in the software (COPD, stroke, IHD, and LC) attributable to PM_2.5_ for 2018. We fed the same input data into AirQ+ to allow for comparability to the results of the main analysis. Regarding the ERF, we selected ‘GBD2015/2016 integrated function 2016′ as the most recent one, which included a fixed counterfactual value of 2.4 µg/m^3^. Regarding the exposure data, AirQ+ only offers to insert one concentration value per area under investigation—in this case, one value for Germany. Therefore, we estimated a population-weighted exposure mean value of 10.8 µg/m^3^ for the year 2018. The formulas from AirQ+ are explained in the manuals of the software, which can be found on t WHO’s website [13].

Within method B, we calculated the EBD based on natural all-cause mortality rates for the year 2018 to compare the results with corresponding estimates from the EEA [5]. For this calculation, we assumed a counterfactual value of 0 µg/m^3^ be comparable with the EEA estimates [5]. We used the RR for natural all-cause mortality as reported by Hoek et al. [14] and the according confidence interval (CI) for the RR to estimate the EBD.

Within a third methodological approach (method C), we quantified the reduction in life expectancy caused by PM_2.5_ exposure for the year 2017, based on the life table approach. The year 2017 was chosen because this was the most recent year with detailed mortality data available. In this approach, life expectancy is re-estimated by assuming that reducing PM_2.5_ concentrations to the level of the counterfactual value leads to lower age-specific death rates [8]. We assumed an annual average PM_2.5_ reduction of 6.5 µg/m^3^ (from 10.7 µg/m^3^ exposure in 2017 to 4.2 µg/m^3^—the counterfactual value used in the main analysis). Life expectancy is then re-estimated with a natural all-cause mortality rate reduced by the effect of PM_2.5_. In this case, we used a RR for natural all-cause mortality reported by Hoek et al. (2013). Spreadsheets provided by the Institute of Occupational Medicine (IOM) were used as a basis for the calculations [15,16]. The models were run for both sexes separately.

### 2.2. PM_2.5_ Exposure Data

For our analysis, we used nationwide annual mean PM_2.5_ concentrations with a spatial resolution of about 2 × 2 km^2^ for the years 2010 to 2018 to estimate the PM_2.5_ exposure in Germany. The PM_2.5_ input data were estimated from a model of nationwide annual mean PM_10_ rural and urban background concentrations using a fixed conversion factor of 0.7 [19], which was validated based on measurement data from Germany.

We combined the calculated nationwide PM_2.5_ concentrations with spatial information on population density (100 × 100 m^2^ resolution) from the 2011 Census [20] to estimate the population-weighted exposure. PM_2.5_ concentrations were then categorized into exposure classes of 1 µg/m^3^ including additional information on the number of people exposed to the respective exposure-class concentrations.

Apart from population health measures such as DALYs, sustainability strategies such as the Global Sustainable Development Goals or the German Sustainable Development Strategy defined further indicators to represent the impacts of air pollutants. These are, for example, the population-weighted annual mean PM_2.5_ exposure, the number of people exposed to concentrations above the WHO air quality guideline value of 10 µg/m^3^ from 2005 [21,22] and of 5 µg/m^3^ from 2021 [23]. The political and public interest for such indicators is increasing. Therefore, we present both indicators.

### 2.3. BoD Input Data

Population data: population data and data on life expectancy for Germany were obtained from the Federal Statistical Office (Wiesbaden, Germany) [24,25]. Both datasets were stratified by five-year age groups and sex.

Mortality data: we used mortality data from the cause-of-death statistics for selected ICD-10 codes as published by the Federal Statistical Office in the German Federal Health Monitoring [26]. The data are stratified by five-year age groups and sex. No garbage-code correction was conducted. Neonatal deaths, as needed for the life table approach, were gathered from the Federal Statistical Office [27].

Morbidity data: there is no official registry for morbidity data in Germany. Therefore, prevalence rates for COPD, IHD, T2DM, and stroke were obtained from the German representative study “German Health Update” (GEDA) [28,29,30,31]. The prevalence rates were stratified by five age groups (18–44, 45–54, 55–64, 65–74, and ≥75 years) and sex. The reference period was 2014/2015. The rates were used to estimate the number of prevalent cases for each year in the time series. For lung cancer, data from the German cancer registry were used to quantify the prevalent cases for each year (2010 to 2015) [32]. At the time of our study, official data were missing for the years 2016 to 2018. Therefore, we used the average prevalence rate of the years 2014 to 2015 as an approximation.

Disability weights (DW): we calculated health outcome, age, and sex-specific DW by dividing the annual YLDs by the prevalence data for five-year age groups and both sexes for each health outcome, year, age group, and sex obtained from the GBD-2019 study data [17]. Using this approach, we were able to adjust the DW for main co-morbidities and to consider different severities within the respective health outcomes. These DW are the same as those from the GBD-2019 study. For an exemplary quantification, please see Table 2.

## 3. Results

### 3.1. PM_2.5_ Exposure in Germany

We estimated the distribution of the German population over the PM_2.5_ exposure categories based on urban and rural background PM_2.5_ concentrations and population density data for the year 2011. Overall, we identified a shift in population exposure toward lower PM_2.5_ annual mean concentrations over time. Figure 1 exemplarily shows a comparison of the years 2010 and 2018: in 2010, the highest share of the population lived in areas with PM_2.5_ concentrations of about 14 µg/m^3^. In 2018, the highest share shifted to concentrations of around 11 µg/m^3^. However, even in 2018, 67% of the population were still exposed to PM_2.5_ concentrations above the WHO air quality guideline value of 10 µg/m^3^ from 2005. Considering the WHO air quality guideline value of 5 µg/m^3^ from 2021 for the year 2018 shows that 99.9% of the German population was exposed to higher concentrations.

The overall trend of decreasing PM_2.5_ exposure in Germany was also reflected in the results of the population-weighted PM_2.5_ annual mean indicator (Figure 2). In 2010, the value was 13.7 µg/m^3^; in 2018 it was considerably lower with 10.8 µg/m^3^, representing a reduction of approximately 21% compared to 2010. However, this trend seems to level off within the last three years of our study.

The EBD was estimated for five health outcomes separately. The detailed results are displayed in Table 3 for the year 2018.

The highest population-attributable fraction (PAF) was estimated for stroke with 11.0% (95% UI 7.8–14.6%), followed by IHD with 10.3% (95% UI 6.5–14.6%). The other PAFs ranged from 6.4% (95% UI 4.5–8.3%) for COPD to 9.8% (95% UI 6.5–12.0%) for T2DM. In absolute numbers, most DALYs were lost due to PM_2.5_ exposure associated with IHD (101,776; 95% UI 62,713–145,644), followed by LC (60,843; 95% UI 43,380–79,379) and T2DM (45,888; 95% UI 30,419–56,481). Likewise, most PM_2.5_-attributable deaths were due to IHD (6977; 95% UI 4285–10,069).

The YLL shares were consistently higher than the YLD shares, for all outcomes except T2DM. It ranged from 64% for COPD to 98% for lung cancer. In contrast, for T2DM the YLL share was significantly lower than the YLD share with 82% of the total disease burden caused by T2DM.

Sex comparisons indicated that men tend to be more strongly affected by the impact of PM_2.5_-related disease burden than women. Compared to women, men’s total attributable burden was much higher for IHD and to a lesser extent also for lung cancer and T2DM. Likewise, their YLLs were generally (slightly) higher. In contrast, women showed higher YLDs for stroke, as well as for COPD, and also more attributable deaths for stroke than men. However, these effects are mainly driven by the underlying burden caused by the different health outcomes and not by the PM_2.5_ exposure.

### 3.2. Time Trend per Health Outcome from 2010 to 2018 in Germany

Figure 3 shows the temporal development of the burden that is due to PM_2.5_ and the five specific health outcomes we considered. Overall, the number of DALYs per 100,000 inhabitants above 25 years decreased between 2010 and 2018 for all health outcomes. The decrease is, however, not as smooth as the exposure decrease (Figure 2). The biggest decrease by about a third can be seen for IHD, followed by stroke. The smallest decrease can be observed for COPD with about 18%. For each health outcome, the year 2012 stands out with a decline in the disease burden, which can be explained by a comparatively lower PM_2.5_ exposure in this year.

### 3.3. Comparison of Our Results with IHME’s GBD-2019 Study Estimates

For most health outcomes, the DALY estimates in the GBD-2019 study [12] were much higher than our estimates of the main analysis (Figure 4). This applies specially to stroke, where the GBD-2019 study estimates were twice as high as the estimates of our main analysis. One reason for the big differences was that the number of recorded cases in the health input data of the GBD-2019 study were much higher than the corresponding input data used in our study. For example, the GBD-2019 study estimated 65,550 stroke deaths per year as compared to 30,975 in the official statistics for Germany [26] and 1,686,220 prevalent cases in contrast to 1,104,900 as derived from the GEDA study [29].

The UI estimated by the GBD-2019 study was much broader compared to the UI we estimated. This does not imply that our estimates were more precise. It rather shows that the GBD-2019 study considered more uncertainties, virtually for each input variable. In contrast, our UIs are only limited to the uncertainty from the RR. The UI of the prevalence data were not used in the quantification and therefore are not displayed in Figure 4.

### 3.4. Method A: AirQ+ Estimates

Assuming that the entire population in Germany is exposed to a mean PM_2.5_ concentration of 10.8 µg/m^3^ in 2018, the attributable burden of disease was quantified with the software AirQ+ for four health outcomes.

For both, PAF and attributable death estimates, the majority of the AirQ+ results were considerably higher than the corresponding results of our main analysis (Figure 5). For example, attributable death estimates were about four times higher for IHD (29,673; 95% UI 12,375–46,342) and about 2.7 times higher for stroke (5082; 95% UI 1268–9625) than the respective results of the main analysis (IHD: 6977; 95% UI 4285–10,069; stroke: 1871; 95% UI 1314–2582). For lung cancer, the results were rather similar with 3445 (95% UI 1558–5900) attributable deaths for AirQ+ as compared to 3785 (95% UI 2699–4939) for our main analysis.

The differences may be explained by the chosen input data and assumptions. For example, in AirQ+ the RR from the GBD-2016 study and a lower counterfactual value of 2.4 µg/m^3^ were used [33]. The lower counterfactual value leads to a higher burden of disease. In addition, a PM_2.5_ mean value was assumed for the total German population, rather than a distribution of the population over different PM_2.5_ exposure classes like in our main analysis.

### 3.5. Method B: Natural All-Cause Mortality

We estimated 687,463 (95% UI 448,887–909,862) YLLs and 58,652 (95% UI 38,298–77,627) deaths from natural all-cause mortality attributable to PM_2.5_ for 2018 in Germany. These figures were comparable to the estimates provided by the EEA (710,900 YLLs and 63,100 deaths) [5]. However, both EEA estimates were basically higher by about 3% and 7%, respectively. These discrepancies most probably are linked to differences in the input data set for PM_2.5_ exposure because EEA used country-specific mortality and life expectancy data, just as we did. Likewise, they used the same RR as recommended by the WHO.

### 3.6. Method C: Reduction in Life Expectancy-Life Table Approach

Using the life table approach, we compared how life expectancy changes when the population is exposed to PM_2.5_ concentrations equal to a counterfactual value of 4.2 µg/m^3^, instead of an annual average concentration of 10.7 µg/m^3^. The results for the life table approach with 2017 as reference year revealed an average reduction in life expectancy at birth of 133 days for men and 118 days for women. This reduces the estimated life expectancy of males from 80.07 to 79.70 years and of females from 84.25 to 83.93 years. These estimates are slightly lower compared to the ones estimated by Apte and Brauer [8] with 0.4 years for the reference year 2016.

### 3.7. Comparison of the Different Methods

One aim of this study was to compare the burden of disease estimates derived by different institutions. Attributable deaths were chosen as the common measure for comparisons, as all institutions provided such estimates. However, the results of the different quantification methods investigated in this study were quite different. In method B and in the EEA estimates, natural all-cause mortality data (ICD-10-codes from A00 to R99) were used, which lead to overall higher estimates of attributable deaths (Figure 6). Therefore, these calculations also included causes of deaths that may not be related to air pollution, which seems to be a likely reason for an overestimation. However, the fact that the investigated health outcomes in this study represent only a selection of attributable health outcomes with the strongest scientific evidence, and recently other studies indicated an increasing number of outcomes being associated with air pollution [10], might decrease this overestimation effect to a certain extent.

The lowest number of attributable deaths for all five investigated health outcomes in total was estimated in the main analysis as 15,652 (95% UI 10,389–21,466). Even though in AirQ+ no deaths that were due to T2DM are quantified (method A), the number of attributable deaths is much higher with 42,469 (95% UI 16,727–69,485). However, the percentage of deaths attributable to T2DM in the main analysis is only 5%.

## 4. Discussion

### 4.1. Overall Decrease in PM_2.5_ Exposure

Based on specific data for Germany, we estimated the disease burden attributable to PM_2.5_ exposure for the years 2010 to 2018. Summarizing the five health outcomes specific to PM_2.5_, the attributable burden resulted in 290,701 (95% UI 194,264–391,111) DALYs for the year 2018. More than a third of the attributable DALYs were due to IHD. From 2010 to 2018, a decline in the attributable burden is visible, as the burden in 2010 was 388,025 (95% UI 265,002–511,499) DALYs. The overall downward trend was strongly related to a decrease in PM_2.5_ exposure over the same period. Accordingly, the population-weighted mean PM_2.5_ exposure and the share of the population living above the 2005 WHO guideline value also decreased. The reduction in PM_2.5_ exposure can be related to the implementation of national and European measures to mitigate particulate matter emissions and their precursors that were introduced to meet the EU limit values as agreed upon in the Ambient Air Quality Directives (EU 2004, 2008). These measures were adopted in national plans to improve ambient air quality and protect human health. In particular, emission mitigation measures for stationary sources (combined heat and power stations, waste incineration plants, and various industrial processes) and the transport sector resulted in overall reduction in PM_2.5_ concentrations in Germany [34]. However, small variations from year to year can be explained by different processes such as changes of weather conditions or economic processes [35,36].

### 4.2. Input Data and Risk Measures

Exposure data: we used a nationwide model for ambient annual mean PM_2.5_ concentrations in combination with census population data to approximate the PM_2.5_ exposure of the German population. A major reason for the difference in the estimates we covered in our study, also in comparison to studies of other institutions, might originate from the characteristics of the underlying input data. In particular, we assumed that differences in the results can be attributed in part to the effect of different spatial resolutions of PM-concentration data.

The model used for PM_2.5_ concentration calculations only represents rural and urban background outdoor air pollution. This limitation very likely led to an underestimation of the exposure and, consequently, the attributable disease burden because PM_2.5_ hot spots, such as urban agglomerations with heavy traffic, were not represented. An adequate representation of these special areas, however, would not only require a different model but also a much finer spatial resolution of the exposure data. The choice of the spatial scale strongly depends on the objective and the desired informative value of the study, as well as on the availability and quality of exposure data. For national and subnational analyses, spatial resolutions of less than 5 × 5 km^2^ are common. For continental or global analyses, resolutions of about 10 × 10 km^2^ are commonly used [37]. To be able to include all countries in the world, the IHME, for example, uses a spatial resolution of 11 × 11 km^2^ for the GBD study [38].

Apart from that, global studies such as the GBD study often use additional PM data sources for exposure estimations to improve the overall model quality, as consistent sources or sufficient data are not always available. These are, for example, satellite data and land-use data, sometimes complemented by topographic information [4,10].

On the European level, the EEA estimated PM_2.5_ concentrations with a comparable spatial resolution at a scale of 10 × 10 km^2^. Since the reporting year 2017, however, the resolution used by EEA has been improved to 1 × 1 km^2^ [39]. The EEA also based its exposure modelling on other data than PM_2.5_ measurements solely by additionally interpolating supplementary measurement data at PM_10_ measurement stations using linear regression and kriging [39].

Several studies confirm that modeled PM_2.5_ concentrations at lower resolutions lead to an underestimation of population exposure and related health impacts [37,40,41]. Exposure models with coarser resolution level do not cover exposure peaks and spatially strongly varying concentrations, which especially occur in urban areas. However, the effect of different spatial resolutions on estimated health impacts is less influential in rural regions, where air pollutant concentrations are often distributed more homogeneously. Depending on the region and considered PM components, though, deviations of model results can vary up to an order of magnitude [37].

Population data: we used the same population density data for Germany to calculate the entire exposure time-series because recent and representative data were not available for all single years, except for 2011. We assumed only minor changes in population distribution and size during the considered time span and, thus, the influence on the EBD results to be negligible.

Finally, to record the actual PM_2.5_ exposure of the population, it would be best to use individual exposure data generated by personal sampling methods. Such detailed data are, however, not yet available in sufficient quality, quantity, and nationwide resolution.

Health data: the number of deaths were obtained from official cause-of-death statistics in Germany and were used without any corrections. This might explain at least some differences in mortality-associated burden when compared to the GBD-2019 study estimates. Many deaths in the German cause-of-death statistics are coded as “heart failure”. Redistributing those deaths to valid codes might result in a substantial increase in deaths to the code group of IHD. In the GBD studies, deaths assigned to the so-called garbage codes or ill-defined codes are redistributed to valid codes, which lead to differences between GBD cause-of-death data and the German cause-of-death statistics of around 27% [42]. This, in turn, also has an impact on the attributable disease burden.

In this study, prevalence rates were taken from representative health surveys conducted in Germany. However, especially for older age groups, these rates were not representative because elderly people tend to be not sufficiently represented in these studies, which could lead to an overestimation of the prevalent cases. Nevertheless, the number of estimated COPD and T2DM cases in the GBD-2019 study were much higher than in our study. We kept the prevalence constant for the considered time span because no continuous data on prevalence were available, which may have resulted in inaccuracies in disease burden, especially for the years more distant from the survey estimates.

We used national sex-specific official life expectancy values for Germany. The life expectancy for women was generally higher than for men. This resulted in more YLLs for females as compared to males. For all deaths at ages above 95 years, the remaining life expectancy at age 90 was applied because no detailed data above the age of 90 years were available. This assumption led to a small overestimation of the disease burden. However, the number of deaths in these older age groups was only small compared to the number in younger age groups. In comparison to the uniform life expectancy used by the GBD-2019 study, the German life expectancy values are lower, which resulted in fewer YLLs in our main analysis.

We quantified the DW by dividing the annual YLDs by the prevalence data for five-year age groups and both sexes as provided by the GBD-2019 study since no country-specific DW were available. Thus, we included the assumption of the GBD-2019 study that everyone has the same DW. However, we used DW adjusted for the severity of the selected diseases stratified by age, sex, and reference year. Using this approach, we were also able to use DW adjusted for comorbidity.

The choice of the RR has a considerable impact on the estimated EBD as Khomenko et al. [43] demonstrated in their study. This may explain the large differences between the results of the main analysis and method B (AirQ+). The RR used in our main analysis was taken from the GBD-2019 study, which included the most recent epidemiological studies at that time [38]. In contrast, AirQ+ provides a selection of RR, which is updated irregularly and even allows the use of customized RR. The RR provided in AirQ+ include studies on active smoking, whereas the RR from the GBD-2019 study excluded active smoking in the ERF derivation. For some health outcomes, e.g., COPD, this leads to lower RR at concentration levels that are typical for the situation in Germany.

To put it in a nutshell, the main limitations and assumptions are listed in the following:PM_2.5_ concentration calculations only represent rural and urban background outdoor air pollution, no hot spots,Factor of 0.7 for PM_10_-to-PM_2.5_ conversion,Constant population density data for Germany for the considered time span,No correction to the official cause-of-death statistics,Constant prevalence rates for the considered time span,Limited representativeness of prevalence rates for oldest age group,We assume the RR and DW are applicable for Germany.

### 4.3. Estimates by Different Institutions

The results of an EBD assessment highly depend on the aim of the study, which in turn affects the assumptions and input data. For instance, the aim of the IHME is to provide comparable and comprehensive data for all countries and a large variety of risk factors, whereas the EEA aims at a comparison of the disease burden that is due to air pollution on the European level. In contrast, our aim was to quantify the EBD attributable to PM_2.5_ with Germany-specific data. AirQ+ is a tool to quantify the disease burden and impact of air pollution for defined regions, populations, and years.

The aim of our study and the aims of the compared studies require different assumptions to be made. For example, for quantifying YLDs in the main analysis, the prevalent approach was applied as also used by the IHME [44]. The IHME uses this approach because the availability of prevalence data is better than that of incidence data, and prevalence data allows a more straightforward comorbidity adjustment [44,45]. In the GBD study, missing data are modeled through complex algorithms like Bayesian belief networks, which allow for filling data gaps and thus for estimating results for every country in the world. On the other side, for countries where data are available, this type of data adjustment can lead to different results, such as different prevalence rates compared to the ones gathered by national representative studies, as shown in our assessment.

The EEA study focuses only on the mortality burden by quantifying the attributable deaths and YLLs that are due to natural all-cause mortality. It relies mainly on previous work done by the WHO and their proposed RR [46], which are, meanwhile, outdated. Nevertheless, for the purpose of the EEA assessment, a comparison of European countries, this approach is acceptable.

With the WHO burden of disease module in AirQ+ also, only the mortality burden can be quantified. Thus, it only represents a part of the total disease burden that is due to PM_2.5_ exposure. The aim of providing a user-friendly software for quantifying the disease burden caused by PM_2.5_ differs from the ones of the IHME GBD-2019 study (aim: providing comparisons between countries and risk factors on a global scale) and the EEA (aim: providing solid estimates for countries on a European scale). In AirQ+, the user can choose different input data and, therefore, control the results to a certain extent. This is especially helpful when estimating the effects of policies or measures in the sense of a health impact assessment, and it is a unique feature of the AirQ+ software.

As a consequence, a direct comparison of the results presented in this article and by the institutions used for comparison is not possible and might lead to skewed interpretations. Each approach and the underlying assumptions can generally be justified with respect to the specific purpose of the study. Ultimately, every burden of disease estimation has to be reviewed and evaluated very carefully in its context regarding the use of input data, risk calculation methods, and resulting estimates.

### 4.4. On the Use of Different Indicators

For quantifying DALYs, several indicators are needed and estimated. These indicators have their own value and meaning. In our study, we quantified the following indicators to present the health effects of PM_2.5_:Population-weighted annual mean PM_2.5_ exposure,Percentage of people exposed to PM_2.5_ levels above WHO guideline value 2005,PAF,YLLs,YLDs,DALYs,Number of attributable deaths,Reduction in life expectancy.

Each indicator has its advantages and disadvantages, which should be considered before using it for communicating the health-related aspects of PM_2.5_. The indicator “percentage of people exposed above the WHO guideline value 2005” is less data-demanding compared to the other indicators. Its interpretation is simple, in particular over time, because only one input parameter, the PM_2.5_ concentration, changes. However, it does not provide any information about the size of the adverse health effects that are due to PM_2.5_ exposure.

The indicator “reduction in life expectancy” is easy to understand because life expectancy is a statistical number that is used in general language. It is widely used even though the computation is complex and requires statistical and demographic expertise. Nevertheless, it is important to argue that the reduction in life expectancy only shows an average time loss for the entire population (or subgroup considered). However, using this information on single individuals would skew the interpretation of the adverse effects of PM_2.5_ because, in reality, there are many people who lose no life years and a few people who in the turn may lose several life years because they die prematurely. Furthermore, this indicator only shows the mortality part of the burden of disease because the burden that is due to time spent with a disease is not considered. Life expectancy indicators are also not flexible over time, and changes of exposure are visible only after long time lags. This makes a timely interpretation of trends complicated.

The number of attributable deaths, often called premature deaths, is a measure often used in public communication. However, in the concept of EBD, all deaths are premature because, statistically, members of each age group still have a remaining life expectancy at the age of death. Therefore, we prefer to use the expression “attributable” as it is more applicable. Furthermore, this measure does not give any information about how much time is lost because of PM_2.5_ exposure, e.g., if children or elderly people died. Thus, it is not an accurate measure, especially when comparing different risk factors among each other.

The most information can be provided by using DALYs because it is a summary measure of population health including both the mortality and morbidity dimensions. On the other hand, this measure is data-demanding, computationally intensive, and difficult to explain, in particular to political stakeholders. In addition, several assumptions are included in this concept, e.g., an optimal life expectancy or the global universality of the DW. Thus, the DALY is not only a descriptive statistical indicator [47].

Depending on the aim of the assessment and especially if the results are used for non-scientific use, e.g., showing the magnitude of the burden caused by air pollution to promote action [48], an appropriate and understandable indicator should be used.

In public communication and political debates, the indicators “percentage of people exposed above WHO guideline value 2005” and “premature deaths” are predominantly used. They include fewer preconditions as underlying assumptions that could be targeted to question their methodologic soundness. However, they only show part of the total burden, as explained before.

The indicator “percentage of people exposed above WHO guideline value 2005” is chosen in the German Sustainable Development Strategy, which states the goal of reducing the share to zero by 2030 [49]. The proposal by the IHME, to use the DALYs as an indicator for the Sustainable Development Goals, was not implemented yet. Consequently, there is a need to present the different indicators and to emphasize the advantages and disadvantages of each.

Authors should discuss the results and how they can be interpreted from the perspective of previous studies. The findings and their implications should be discussed in the broadest context possible. Future research directions may also be highlighted.

## 5. Conclusions

We could show that the amount of DALYs, YLLs, or YLDs, or number of attributable deaths estimated with the burden of disease approach highly depend on the underlying assumptions and input data and, thus, also on the purpose of the respective assessments. More transparency is needed to explain why certain input data are used and why results may differ from other studies. This is especially true for PM_2.5_ because the evidence base concerning the RR is evolving quickly and the impact of the applied RR on the resulting burden is considerable. The indicators estimated with the burden of disease method are not simple objective statistics or measures. However, they can be used to demonstrate the impact of risk factors on population health in a quantitative way and, by that means, they can significantly contribute to discussions and decisions in the field of environmental health issues.

## Figures and Tables

**Figure 1 ijerph-19-13197-f001:**
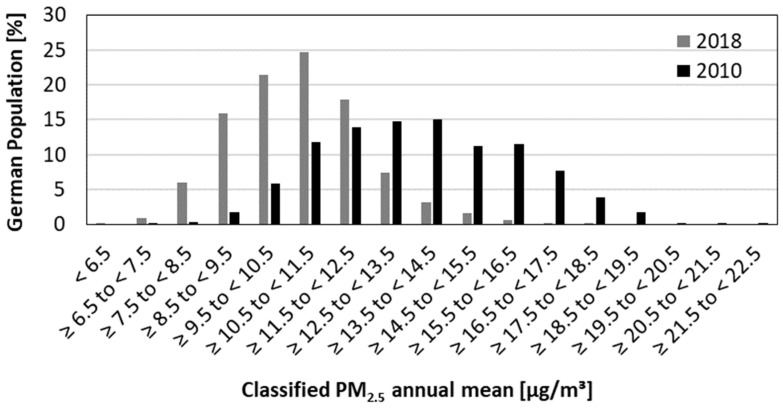
Frequency distribution of PM_2.5_ annual mean exposure and the German population for 2010 and 2018.

**Figure 2 ijerph-19-13197-f002:**
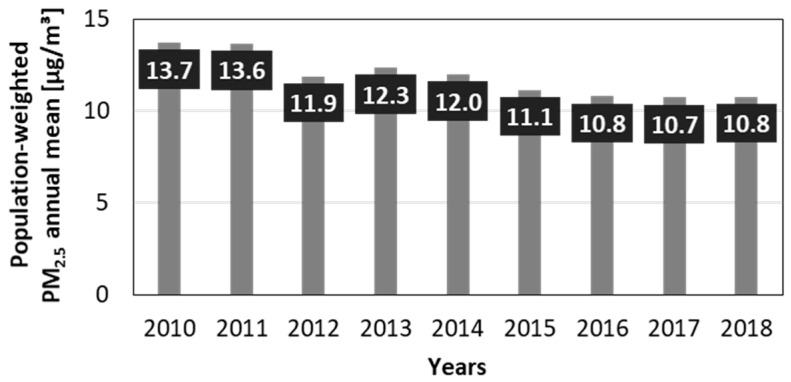
Nationwide population-weighted PM_2.5_ annual mean exposure from 2010 to 2018 in Germany.

**Figure 3 ijerph-19-13197-f003:**
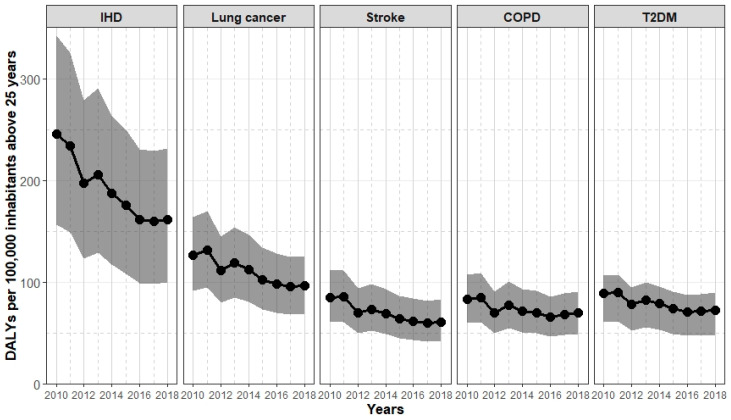
Burden of disease (DALYs per 100,000 people above 25 years including 95% UI) from 2010 to 2018.

**Figure 4 ijerph-19-13197-f004:**
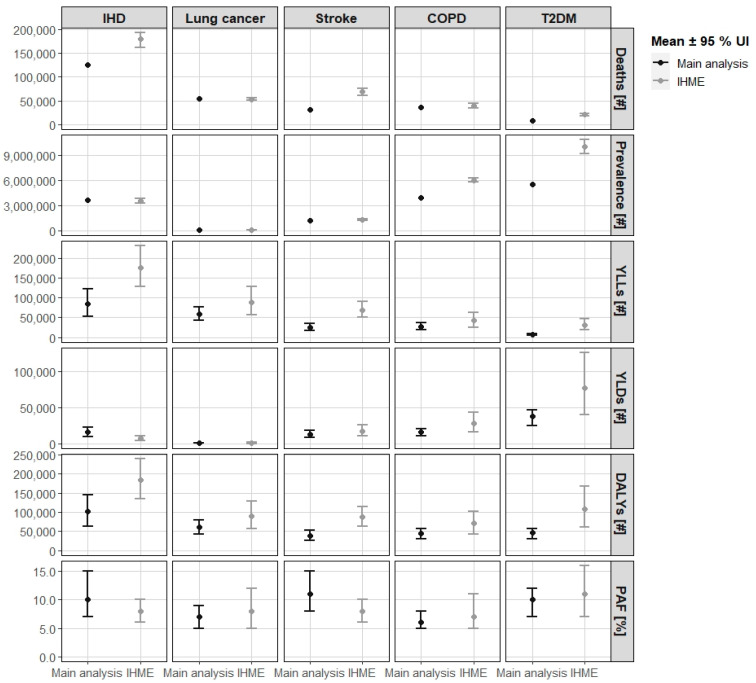
Comparison of the main analysis with IHME’s GBD-2019 study estimates for the year 2018. PAF = population attributable fraction. YLLs = years of life lost due to mortality. YLDs = years lived with disability. DALYs = disability-adjusted life years. UI = uncertainty interval. COPD = chronic obstructive lung disease. GBD = global burden of disease. IHD = ischemic heart disease. IHME = Institute for Health Metrics and Evaluation. T2DM = type 2 diabetes mellitus.

**Figure 5 ijerph-19-13197-f005:**
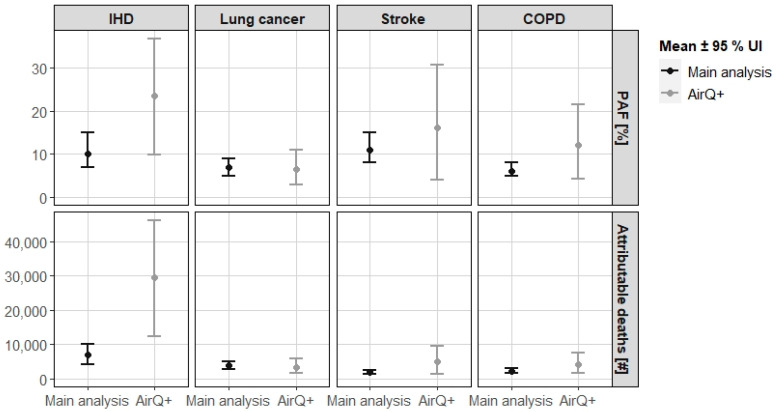
Comparison of the main analysis with estimates generated with AirQ+. COPD = chronic obstructive lung disease. IHD = ischemic heart disease. PAF = population attributable fraction. UI = uncertainty interval.

**Figure 6 ijerph-19-13197-f006:**
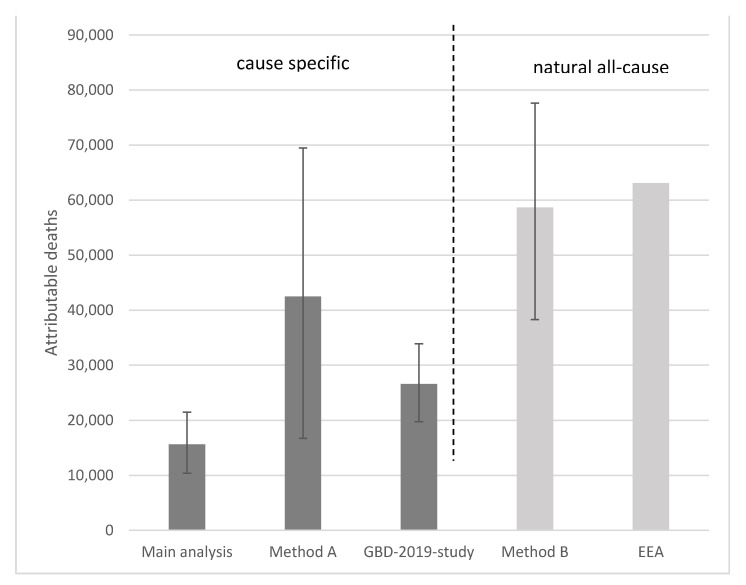
Attributable deaths that are due to PM_2.5_ exposure in Germany 2018.

**Table 1 ijerph-19-13197-t001:** Overview of the different analyses and approaches used.

Analysis Name	Approach	Health Outcomes (Number)	Relative Risk	Indicator(s)	Target Year(s)	Software Used	Comparison
Main analysis	EBD incl. PAF	Cause-specific (5): COPD, stroke, IHD, LC, T2DM	[17]	YLLs, YLDs	2010–2018	Excel	[17], Method A
Method A	EBD incl. PAF	Cause-specific (4): COPD, stroke, IHD, LC	IHME as provided in AirQ+ *	YLLs	2018	AirQ+ (WHO)	Main analysis
Method B	EBD incl. PAF	Natural all-cause mortality	[14]	YLLs	2018	Excel	[5]
Method C	Life table	Natural all-cause mortality	[14]	Life expectancy reduction in days	2017	Excel	[8]

EBD = environmental burden of disease. PAF = population attributable fraction. COPD = chronic obstructive lung disease. IHD = ischemic heart disease. LC = lung cancer. T2DM = type 2 diabetes mellitus. YLLs = years of life lost due to mortality. YLDs = years lived with disability. EEA = European Environment Agency. IHME = Institute for Health Metrics and Evaluation. WHO = World Health Organization. * Software provided by the WHO Regional Office for Europe, European Centre for Environment and Health [18].

**Table 2 ijerph-19-13197-t002:** Exemplary DW quantification for IHD and the year 2017.

	YLD	YLD	Prevalence	Prevalence	Disability Weight	Disability Weight
Age	Male	Female	Male	Female	Male	Female
**<1 year**	0	0	0	0	0.000	0.000
**1 to 4**	0	0	0	0	0.000	0.000
**5 to 9**	0	0	0	0	0.000	0.000
**10 to 14**	0	0	0	0	0.000	0.000
**15 to 19**	12	9	273	216	0.045	0.042
**20 to 24**	54	44	1069	822	0.051	0.054
**25 to 29**	142	104	2732	1991	0.052	0.052
**30 to 34**	263	194	5653	3747	0.047	0.052
**35 to 39**	460	330	11,914	7093	0.039	0.047
**40 to 44**	863	589	24,036	13,238	0.036	0.044
**45 to 49**	1950	1226	55,432	28,204	0.035	0.043
**50 to 54**	3495	2084	104,267	49,695	0.034	0.042
**55 to 59**	4797	2743	147,306	68,581	0.033	0.040
**60 to 64**	6185	3423	181,049	85,943	0.034	0.040
**65 to 69**	8218	4735	230,099	116,939	0.036	0.040
**70 to 74**	9201	5683	240,107	137,598	0.038	0.041
**75 to 79**	12,692	8735	326,060	217,370	0.039	0.040
**80 to 84**	9736	7731	235,017	194,260	0.041	0.040
**85 to 89**	5155	5077	115,685	130,013	0.045	0.039
**90 to 94**	1853	2373	38,284	61,610	0.048	0.039
**95 plus**	455	661	7387	15,436	0.062	0.043

**Table 3 ijerph-19-13197-t003:** PAF, YLLs, YLDs, DALYs, and attributable deaths per health outcome and sex attributable to long-term PM_2.5_ exposure in 2018 (95% UI in brackets).

		PAF	YLLs	YLDs	DALYs	Attributable Deaths
IHD	Sum	10.3% (6.5–14.6%)	85,483 (52,676–122,380)	16,293 (10,037–23,264)	101,776 (62,713–145,644)	6977 (4285–10,069)
	Male		56,189 (34,675–80,219)	9774 (6033–13,934)	65,962 (40,708–94,152)	4094 (2515–5887)
	Female		29,294 (18,001–42,162)	6519 (4005–9330)	35,814 (22,005–51,492)	2882 (1770–4181)
	Rate per 100,000		135.63 (83.58–194.17)	25.85 (15.93–36.91)	161.48 (99.50–231.08)	11.07 (6.80–15.97)
Lung cancer	Sum	7.1% (5.0–9.2%)	59,487 (42,414–77,611)	1355 (966–1768)	60,843 (43,380–79,379)	3785 (2699–4939)
	Male		34,450 (24,562–44,945)	896 (639–1169)	35,346 (25,201–46,115)	2297 (1638–2997)
	Female		25,037 (17,851–32,666)	459 (327–599)	25,497 (18,179–33,265)	1488 (1061–1942)
	Rate per 100,000		94.38 (67.29–123.14)	2.15 (1.53–2.81)	96.53 (68.83–125.94)	6.01 (4.28–7.84)
Stroke	Sum	11.0% (7.8–14.6%)	25,019 (17,579–34,225)	13,399 (9396–18,365)	38,417 (26,975–52,590)	1871 (1314–2582)
	Male		12,533 (8804–17,117)	5917 (4150–8098)	18,450 (12,955–25,215)	891 (625–1228)
	Female		12,486 (8775–17,107)	7481 (5245–10,267)	19,967 (14,020–27,375)	980 (689–1354)
	Rate per 100,000		36.69 (27.89–54.30)	21.26 (14.91–29.14)	60.95 (42.80–83.44)	2.97 (2.08–4.10)
COPD	Sum	6.4% (4.5–8.3%)	27,847 (19,578–36,269)	15,931 (11,200–20,749)	43,777 (30,777–57,017)	2241 (1588–2919)
	Male		14,680 (10,321–19,120)	7620 (5357–9925)	22,300 (15,678–29,044)	1219 (857–1588)
	Female		13,167 (9257–17,149)	8310 (5843–10,824)	21,477 (15,099–27,973)	1022 (719–1331)
	Rate per 100,000		44.18 (31.06–57.54)	25.28 (17.77–32.92)	69.46 (48.83–90.46)	3.56 (2.50–4.63)
T2DM	Sum	9.8% (6.5–12.0%)	8128 (5388–10,004)	37,760 (25,031–46,477)	45,888 (30,419–56,481)	778 (516–957)
	Male		4196 (2781–5164)	20,596 (13,653–25,350)	24,791 (16,434–30,515)	364 (241–448)
	Female		3932 (2607–4840)	17,164 (11,378–21,127)	21,096 (13,985–25,967)	414 (274–509)
	Rate per 100,000		12.90 (8.55–15.87)	59.91 (39.71–73.74)	65.08 (48.26–89.61)	1.23 (0.82–1.52)

The attributable disease burden was estimated for the German population aged above 25 years. PAF = population attributable fraction. YLLs = years of life lost due to mortality. YLDs = years lived with disability. DALYs = disability-adjusted life years. UI = uncertainty interval. COPD = chronic obstructive lung disease. IHD = ischemic heart disease. LC = lung cancer. T2DM = type 2 diabetes mellitus.

## Data Availability

The datasets analyzed during the current study are available from the corresponding authors upon reasonable request.

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
