# Peer review of "Burden of Disease Due to Ambient Particulate Matter in Germany—Explaining the Differences in the Available Estimates"

_ijerph, 2022, doi:10.3390/ijerph192013197_

Round 1
Reviewer 1 Report
Tobollik et al. conducted a comparative analysis across methodological approaches developed by themselves and those implemented by reference international institutions (IHME; WHO; EEA) to estimate the environmental burden of disease (EBD) applied to Germany (2010-2018), highlighting similarities and discrepancies in the input data, computational steps, and assumptions as well as their impact on estimation.
Major comments:
Methods.
1. Considering the importance of the years lived with disability (YLDs) to the estimation of Disability-adjusted life years (DALYs), and particularly, because the authors calculate disability weights (DW) from their own data, the section would benefit from a clear explanation of how YLD were derived. The authors might include such methods to the supplementary material (appendix).
Results.
1. The authors should avoid discussing their results within the same section as these are presented. I suggest moving those comments to the discussion section to avoid duplication or redundancy in the manuscript. For example, the authors reported their findings in lines 218-220 ("For most health outcomes the DALYs estimates in the GBD-2019-study [11] were much higher than our estimates (Figure 4). This applies specially to stroke, where the GBD-2019-study estimates were twice as high as the estimates of our main analysis."), followed by a commentary to these results in lines 220-222 ("One reason for the big differences probably was that the number of recorded cases in the health input data of the GBD-2019-study were much higher than the corresponding input data used in our study.").
2. Lines 204-205: Here is another example of discussing results within the same section. In addition, no finding from the analysis or citation supports the statement "However, these effects are driven by the underlying burden due to the different health outcomes and not PM2.5 exposure".
3. Figure 4: Why don't prevalence estimates from the GEDA have uncertainty/confidence limits if they were derived from a sample of the German population? or they do have limits, but these are so precise that can't be seen in the figure.
Minor comments:
Methods.
1. Line 80: The authors state that "A sample spreadsheet can be found in the appendix.", however, there is no such supplementary material in the submission.
Results.
1. Figure 1: Since the Y-axis does not represent a cumulative distribution, labels of the X-axis should include low and high bounds (e.g., >=9.5 to <10.5, etc.).
2. Line 180. Replace Table 1 by Table 2.
Conclusions.
1. Line 512: It seems "6. Patents" at the end of the line is a typo.
Author Response
Answers to reviewer 1:
Thank you very much for your constructive feedback, which we tried to implement as much as possible. The comments helped us a lot to improve the manuscript and to make it more understandable.
Please find our answers to your comments below:
Comment: Methods. 1. Considering the importance of the years lived with disability (YLDs) to the estimation of Disability-adjusted life years (DALYs), and particularly, because the authors calculate disability weights (DW) from their own data, the section would benefit from a clear explanation of how YLD were derived. The authors might include such methods to the supplementary material (appendix).
Answer: Thanks for the comment. We consider the YLDs as an important part of the DALY, because only with the YLDs the DALYs can be considered a summary measure of population health. We did not calculate disability weights based on newly generated data. In fact, the data originate from the Global Burden of Disease-2019-Study. We requested disability weights from the Global Burden of Disease-2019-Study, but it was not possible to get them in the detail we needed them for our quantification. Therefore, we got the answer by IHME that by just dividing the annual YLDs by the prevalence data, we get the fully adjusted disability weights used in the Global Burden of Disease-2019-Study. To be more transparent, we included an example table in the article and the following sentence: “These DW are the same such as those used in the GBD-2019 study.“
Comment: Results. 1. The authors should avoid discussing their results within the same section as these are presented. I suggest moving those comments to the discussion section to avoid duplication or redundancy in the manuscript. For example, the authors reported their findings in lines 218-220 ("For most health outcomes the DALYs estimates in the GBD-2019-study [11] were much higher than our estimates (Figure 4). This applies specially to stroke, where the GBD-2019-study estimates were twice as high as the estimates of our main analysis."), followed by a commentary to these results in lines 220-222 ("One reason for the big differences probably was that the number of recorded cases in the health input data of the GBD-2019-study were much higher than the corresponding input data used in our study.").
- Lines 204-205: Here is another example of discussing results within the same section. In addition, no finding from the analysis or citation supports the statement "However, these effects are driven by the underlying burden due to the different health outcomes and not PM2.5 exposure".
Answer: Thank you very much for your comment. We discussed this aspect prior to submitting the article. Because, as you are correctly pointing out it is not common to discuss the results in the results section. The reason for including such additional information was that the focus of the article, as displayed in the title, is not only to show the results but also to explain the differences in the estimates generated by different approaches. We feel that it would rather be less readable if we would first show the detailed results and then repeat the results in the discussion for the single comparisons. We believe that the way we have structured our results and discussion section is more efficient and suitable. If you still see the need for a restructuring, please let us know. We will then do it.
Comment: 3. Figure 4: Why don't prevalence estimates from the GEDA have uncertainty/confidence limits if they were derived from a sample of the German population? or they do have limits, but these are so precise that can't be seen in the figure.
Answer: The prevalence estimates from GEDA present confidence intervals. But we did not use them in our quantifications. Thus, we also did not include them in the figure to avoid misunderstandings. We have added a sentence to clarify this issue in the main text: “The UI of the prevalence data were not used in the quantification and therefore are not displayed in figure 4”.
Comment: Methods.1. Line 80: The authors state that "A sample spreadsheet can be found in the appendix.", however, there is no such supplementary material in the submission.
Answer: Thank you very much for the comment. We now added the supplementary material.
Comment: Results. 1. Figure 1: Since the Y-axis does not represent a cumulative distribution, labels of the X-axis should include low and high bounds (e.g., >=9.5 to <10.5, etc.).
Answer: Thanks a lot, we changed it accordingly.
Comment: 2. Line 180. Replace Table 1 by Table 2.
Answer: Thanks a lot. We changed it in table 3 to ensure a consecutive order of the tables.
Comment: 1. Line 512: It seems "6. Patents" at the end of the line is a typo.
Answer: Thanks a lot. It was deleted.

Reviewer 2 Report
In the present manuscript, Tobollik et al. estimate PM2.5-associated burden of disease in Germany for 2010-2018 and compare with already available approaches including WHO´s environmental burden of disease (EBD) and AirQ+ software. In some cases, significant discrepancies were obtained when comparing with estimations from other sources (line 217-229). Unfortunately, numerous assumptions (e.g. PM2.5 data from PM10 data+0.7 factor, line 121) and limitations (e.g. missing data, line 150) significantly weaken the argument. In addition, the discussion section is rather ramble, specially sub section “on the use of different indicators”.
Author Response
Answer to reviewer 2:
Comment: In the present manuscript, Tobollik et al. estimate PM2.5-associated burden of disease in Germany for 2010-2018 and compare with already available approaches including WHO´s environmental burden of disease (EBD) and AirQ+ software. In some cases, significant discrepancies were obtained when comparing with estimations from other sources (line 217-229). Unfortunately, numerous assumptions (e.g. PM2.5 data from PM10 data+0.7 factor, line 121) and limitations (e.g. missing data, line 150) significantly weaken the argument. In addition, the discussion section is rather ramble, specially sub section “on the use of different indicators”.
Answer: Thank you for your comment. It is common to base environmental burden of disease estimates on assumptions. In our article we described the assumptions and explained the impact of these on the results. The assumptions are based on established methods, such as the conversion factor of 0.7. This factor is recommended by the WHO. Missing data is as well very common for (environmental) burden of disease estimates. If you compare our estimations to the ones of the Global Burden of Disease study, we already used an improved data base.
The argument is not weakened in our view, because the other institutions base their estimates on assumptions as well. We discuss the impact of the assumptions in very detail to ensure a transparent interpretation of the results.
The section “on the use of different indicators” goes beyond the scope of a simple burden of disease quantification. However, based on our experience with health-related indicators and how they are used and communicated, we want to emphasize that it is important to be aware what an indicator is able to present and what not.
If the reviewer has more specific comments, we are very happy to implement these.

Reviewer 3 Report
In the manuscript "Burden of disease due to ambient particulate matter in Germany - explaining the difference", the authors apply different methods for calculation of the burden of disease (e.g. number of years of life lost) due to PM2.5 exposure. Air pollution causes a major threat to human health, and is therefore an important topic of interest.
However, the description of the methods used is not sufficiently transparent to reproduce the results. The authors mention software used for the calculations of the outcome indicators, e.g. Excel and AirQ+, but there is no mention of the equations used within these software packages. The discussion of why these methods are selected and why the results differ is also rather limited, while this seems to be the main objective of the study. A thorough revision is needed to make the methods more clear and reproducible, including equations on how the indicators are calculated and based on what input data.
Detailed comments:
- The title of the manuscript is unclear. "explaining the difference": difference in what? The title should be readable independently from the rest of the manuscript.
- Variant A, B, C: the use of "variant" throughout the text can be confusing for a reader in epidemiology, which might have a different meaning of the word in mind. Would something like Method A, B, C be a better wording?
- Please check the manuscript of spelling and editorial errors throughout (e.g. line 13, line 206, line 512).
Author Response
Answers to reviewer 3:
Thank you very much for your constructive feedback, which we tried to implement as much as possible. The comments helped us a lot to improve the article and to make it more understandable.
Please find our answers to your comments below:
Comment: However, the description of the methods used is not sufficiently transparent to reproduce the results. The authors mention software used for the calculations of the outcome indicators, e.g. Excel and AirQ+, but there is no mention of the equations used within these software packages. The discussion of why these methods are selected and why the results differ is also rather limited, while this seems to be the main objective of the study. A thorough revision is needed to make the methods more clear and reproducible, including equations on how the indicators are calculated and based on what input data.
Answer: Thank you very much for your comment. We added a supplementary file which includes our entire quantification process. All equations can be found there. The formulas from AirQ+ are explained in the manuals of the software which can be found here: https://www.who.int/europe/tools-and-toolkits/airq---software-tool-for-health-risk-assessment-of-air-pollution
We also went through the method section and added explanations why we used certain methods . The explanation of the reasons for the differences in the results can be found in the results section.
Comment: The title of the manuscript is unclear. "explaining the difference": difference in what? The title should be readable independently from the rest of the manuscript.
Answer: Thanks a lot for the comment. The heading was adjusted. It is now: “Burden of disease due to ambient particulate matter in Ger-many – explaining the differences in the available estimates”
Comment: Variant A, B, C: the use of "variant" throughout the text can be confusing for a reader in epidemiology, which might have a different meaning of the word in mind. Would something like Method A, B, C be a better wording?
Answer: Thanks a lot for your comment. We changed the wording to method.
Comment: Please check the manuscript of spelling and editorial errors throughout (e.g. line 13, line 206, line 512).
Answer: Thanks a lot. We have checked the document for spelling mistakes.

Reviewer 4 Report
Comments on the manuscript entitled “Burden of disease due to ambient particulate matter in Germany – explaining the difference”
The authors conducted this study to estimate the environmental burden of disease (EBD) for the German population associated with PM2.5-exposure in Germany from 2010 to 2018. In this manuscript, the research methods are adequately described in detail and the results are discussed in depth. The authors addressed that a transparent description of input data, computational steps, and assumptions is essential to explain differing results of EBD studies to improve methodological credibility and trust in the results. The reviewer strongly agrees with this conclusion. The manuscript was well written and properly organize. One of the limitations was that the input data of PM2.5 concentration was obtained from simple estimation based on PM10, not from real observation. Despite the limitation, this study is still valuable for future related researches. In summary, my suggestion is “acceptance after minor revision”
Specific comments:
1. Line 7 and the full text. It should be “PM2.5”. “2.5” should be shown as subscript.
2. Line 21. I guess it should be “with caution”, not “which caution”.
3. Line 131-132. The latest WHO Air Quality Guideline value for PM2.5 was lowered from 10 μg/m3 to 5 μg/m3, which was released in September 2021.
4. Line 74-75. What is the basis for setting the counterfactual value of 4.2 µg/m3? Why not set it as WHO Air Quality Guideline value for PM2.5?
5. Line 93. I don't understand why the author assumed a counterfactual value of 0 µg/m3, which seems unreasonable because PM2.5 in the ambient air cannot be reduced to 0 µg/m3.
6. Line 180. It should be Table 2, not Table 1.
7. Line 199-Line 204. From the results, sex-comparisons indicated that men tend to be more strongly affected by the impact of PM2.5-related disease burden than women. What is the possible reasons?
8. Line 206. Format adjustment is needed.
9. Line 512. “6. Patents” is irrelevant, which may be deleted.
Author Response
Answers to reviewer 4:
Thank you very much for your constructive feedback, which we tried to implement as much as possible. The comments helped us a lot to improve the article and to make it more understandable.
Please find our answers to your comments below:
Comment: 1. Line 7 and the full text. It should be “PM2.5”. “2.5” should be shown as subscript.
Answer: Thanks for the comment. We made the changes accordingly.
Comment: 2. Line 21. I guess it should be “with caution”, not “which caution”.
Answer: Thanks a lot for the comment. We corrected the typo.
Comment: 3. Line 131-132. The latest WHO Air Quality Guideline value for PM2.5 was lowered from 10 μg/m3 to 5 μg/m3, which was released in September 2021.
Answer: Thanks a lot. The WHO AQG2021 were included in the methods and the results section.
Comment: 4. Line 74-75. What is the basis for setting the counterfactual value of 4.2 µg/m3? Why not set it as WHO Air Quality Guideline value for PM2.5?
Answer: The goal of the WHO AQG is to recommend a numerical value below which no or only small negative health effects are expected in order to guide standards. We used a counterfactual value, which represents the lowest observed concentration in epidemiological studies from which the RR is taken. According to the WHO also health effects below a mean annual exposure of 5 µg/m3 are possible. Thus, we refer to a value which is in line with the RR, which we take for your estimation. To make it clearer we added the following sentence: “... which base on the same epidemiological studies from which the relative risks (RR) are taken from”. We are also in line with the current recommendations of the GBD 2019 study, which also uses 4.2 µg/m3 as the counterfactual value.
Comment: 5. Line 93. I don't understand why the author assumed a counterfactual value of 0 µg/m3, which seems unreasonable because PM2.5 in the ambient air cannot be reduced to 0 µg/m3.
Answer: We fully agree. 0 µg/m3 PM2.5 is not achievable due to natural sources. In this scenario analysis it was not our goal to estimate the realistic burden of disease. The main aim was to replicate the quantifications of the EEA to compare our results to their estimates. Therefore, we had to use the same assumptions.
Comment: 6. Line 180. It should be Table 2, not Table 1.
Answer: Thanks, it was corrected.
Comment: 7. Line 199-Line 204. From the results, sex-comparisons indicated that men tend to be more strongly affected by the impact of PM2.5-related disease burden than women. What is the possible reasons?
Answer: The main reason is explained in the last sentence: “However, these effects are driven by the underlying burden due to the different health outcomes and not PM2.5 exposure.” Thus, it is not the PM2.5 exposure (which is the same for both sexes) which leads to the differences. As an example, men die earlier due to most of considered diseases.
Comment: 8. Line 206. Format adjustment is needed.
Answer: Thanks a lot. It was changed accordingly.
Comment: 9. Line 512. “6. Patents” is irrelevant, which may be deleted.
Answer: Thanks a lot. This word was deleted.

Round 2
Reviewer 2 Report
This reviewer understands the estimations and assumptions needed to perform burden of disease studies. However, while they are described in the results section, the manuscript will benefit from a separated section on the conclusion section that summarizes and highlights these limitations. If the authors are still willing to keep the “On the use of different indicators” subsection, this reviewer suggest to make it more concise and avoid using vague terms and expressions (e.g.: “relatively easy to understand”, line 469; “quite complex”, line 470; “quite important”, line 471; “not very flexible”, line 478, and so).
Author Response
Dear Reviewer,
thank you very much for responding to our request and making more explicit comments. They helped us to improve the manuscript and to make it more understandable.
We included a section summarizing the limitations and assumptions. They are listed in a short but comprehensive way in bullet points at the end of the section “Input data and risk measures” as follows:
To put it in a nutshell the main limitations and assumptions are listed in the following:
- PM2.5-concentration calculations only represents rural and urban background outdoor air pollution, no hot spots
- Constant population density data for Germany for the considered time span
- Conversion factor of 0.7 for PM10 to PM2.5 conversion
- No correction to the official cause-of-death statistics
- Constant prevalence rates for the considered time span
- Limited representative of prevalence rates for oldest age group
- We assume the RR and DW are applicable for Germany
Thank you very much for your comments to the indicators sections. We agree these vague expressions are not helpful to the reader. We deleted these and specified the wording where we saw it as appropriated, for example the comprehensibility of life expectancy as an indicator.
Reviewer 3 Report
I would like to thank the authors for their response to my comments. The authors have addressed my concerns and improved the manuscript in their resubmitted version.
In the response letter, the authors mention " The formulas from AirQ+ are explained in the manuals of the software which can be found here: https://www.who.int/europe/tools-and-toolkits/airq---software-tool-for-health-risk-assessment-of-air-pollution". However, I do not find this information and link in the manuscript (or maybe I have missed it because the response letter does not include a reference to the manuscript text pointing towards the line numbers that have changed - a suggestion for future responses). Please add this information and reference to the manuscript.
Author Response
Dear Reviewer,
thank you very much for your answer and comment. Indeed, this explicit referencing to the formula and the manuals was missing in the manuscript. Therefore, we added, as you recommended, the sentence: “The formulas from AirQ+ are explained in the manuals of the software which can be found here [13]. Under 13, the link to the WHO homepage with the manuals and the software itself can be found